# Synthetic Data Generation to Speed-Up the Object Recognition Pipeline



**Damiano Perri [1,2,*]**, **Marco Simonetti [1,2]** and **Osvaldo Gervasi [2]**

1 Department of Mathematics and Computer Science, University of Florence, 50134 Florence, Italy; m.simonetti@unifi.it
2 Department of Mathematics and Computer Science, University of Perugia, 06123 Perugia, Italy; osvaldo.gervasi@unipg.it
* Correspondence: damiano.perri@unifi.it

**Abstract:** This paper provides a methodology for the production of synthetic images for training neural networks to recognise shapes and objects. There are many scenarios in which it is difficult, expensive and even dangerous to produce a set of images that is satisfactory for the training of a neural network. The development of 3D modelling software has nowadays reached such a level of realism and ease of use that it seemed natural to explore this innovative path and to give an answer regarding the reliability of this method that bases the training of the neural network on synthetic images. The results obtained in the two proposed use cases, that of the recognition of a pictorial style and that of the recognition of men at sea, lead us to support the validity of the approach, provided that the work is conducted in a very scrupulous and rigorous manner, exploiting the full potential of the modelling software. The code produced, which automatically generates the transformations necessary for the data augmentation of each image, and the generation of random environmental conditions in the case of Blender and Unity3D software, is available under the GPL licence on GitHub. The results obtained lead us to affirm that through the good practices presented in the article, we have defined a simple, reliable, economic and safe method to feed the training phase of a neural network dedicated to the recognition of objects and features to be applied to various contexts.

**Keywords:** Unity3D; Blender; virtual reality; synthetic dataset generation; machine learning; neural networks

## 1. Introduction

Machine learning is now one of the areas where scientific research is focused most. To train the neural networks correctly, it is necessary to have available datasets of examples that the network can use to learn and understand how to solve the problem that is assigned to it. The datasets for the training of the neural networks are generally very large and require considerable efforts to be constructed correctly. Let us think, for example, of the convolutional neural networks: these are used today to extract the features that compose images and classify them according to the labels that the programmer has predefined. As an example, let us imagine a dataset for the binary classification of animals, such as dogs or cats. Unless we consider a dataset that is already available on the web, it is necessary to create one specific to the problem to be addressed. Such a dataset would be very complex and expensive to create in the real world and is a general case of representing three-dimensional environments with completely random lighting conditions and object arrangements, so it may be appropriate and advantageous to create it virtually, thanks to the enormous developments that have taken place in 3D modelling software. This solution allows us to recreate virtual scenarios, generating a high number of images with specific techniques of scene illumination and a random arrangement of objects to enrich the amount of information to be fed to the neural network for its training. In this

paper, we present a technique for generating synthetic datasets, using the popular three-dimensional environment modelling software Unity3D (https://unity.com/ (accessed on 15 November 2021)) and Blender (https://blender.org/ (accessed on 15 November 2021)). We illustrate two different use cases, which we think can be of valuable help to researchers, as they are general cases that can help to solve specific problems. The first *use case* involves generating a dataset for the binary classification of paintings of the Baroque and Impressionist styles. With this example, we want to show and describe how it is possible to generate images of objects (in our case paintings) and how they can be analysed using neural networks. This use case, therefore, involves framing a three-dimensional model from various angles and under various lighting conditions. The second use case involves generating a dataset for recognising men at sea, specifically migrants. This example is completely different from the previous one because instead of having an object, we have a scenario, and through graphical modelling environments, we can easily recreate different weather and light conditions: for example, we can have a scenario with the typical sunlight of the morning, afternoon, evening or night. We can also simulate what men at sea would look like in different weather conditions, for example, clear skies or cloudy skies. A further qualifying aspect of our work is that we have automated the process of data augmentation on virtually generated images by manipulating each image in various aspects. The sections that make up this paper are divided in the following manner. In Section 2, we analyse articles and manuscripts that have dealt with this problem or have analysed the correlations that these methodologies may have with the world of scientific research. Section 3 describes how we organised the research, analysing both the theoretical and practical development of our work. In Section 4, we describe our results and also analyse best practices for generating synthetic datasets, using three-dimensional environment modelling software. Section 5 describes the techniques we recommend for generating synthetic datasets. Section 6 reports the conclusions we drew as a result of the experimental analysis described in this paper. The datasets that are generated for this article and the codes we developed to implement them were made public and are freely accessible via the GitHub page (https://github.com/DamianoP/DatasetGenerator (accessed on 15 November 2021)). As shown in the Supplementary materials. All the code on the repository is open source, licensed under the GNU General Public License v3.0, and freely usable by anyone.

## 2. Related Works

Supervised machine learning for image recognition, like any other human endeavour, has yielded significant advantages while simultaneously posing new challenges. Indeed, one of the most significant concerns picture recognition reliability when the data collection comprises a small number of samples on which to base training.

Although there are several picture databases online, both private and public, even providing open access [i.e., ImageNet (https://www.image-net.org (accessed on 7 October 2021)), OpenImages (https://github.com/openimages (accessed on 7 October 2021)), SUN database (http://vision.princeton.edu/projects/2010/SUN/SUN397.tar.gz (accessed on 7 October 2021)), Microsoft Common Objects in Context—COCO (https://cocodataset.org (accessed on 7 October 2021)), PASCAL VOC dataset (http://host.robots.ox.ac.uk/pascal/VOC/databases.html (accessed on 7 October 2021)), and OpenLORIS-Object (https://lifelong-robotic-vision.github.io/dataset/ (accessed on 7 October 2021))], there are still many niches of themes that are not depicted or for which there are only a few photos accessible. So, many techniques were devised to enhance the number of available occurrences, such as geometric transformations [1–3], parameter adjustment (number of pixels, colour mappings, contrast, multi-spectral bands, etc.) [4,5], and synthetic picture generation [via GAN/RAN [6–10], or graphics engines, such as Unity3D, GODOT (https://godotengine.org/ (accessed on 7 October 2021)) [11], and Unreal (https://www.unrealengine.com (accessed on 7 October 2021))].

Another problem deriving from the use of some particular types of datasets is the imbalance of the classes in the classification process (*unbalanced classes*) [12,13]. This phe-

nomenon was shown to have a negative impact on traditional classifier training. Many of these strategies can efficiently solve the problem of unbalancing classes [14–17] in datasets when their quantity is not exactly uniform, but oversampling minor classes [18–20], undersampling major ones [21–23], and the possibility of weighing the network parameters in a different way so as to appropriately rebalance all the classes [24–26] appear to be winning strategies too. By contrast, this issue appears to be decreasing in all binary classification methods [27,28].

Furthermore, synthetic scenario creation is becoming increasingly relevant in a variety of fields [29–35], ranging from an object and figure identification and categorisation to automated recognition and tracing [36–39]. The adaptability and reusability of this method is exemplified by the ability to train networks to detect specific items or people in very challenging circumstances. So, the use of powerful graphics engines that are able to reproduce reality, or a scenario to be represented, in a very realistic way is, therefore, becoming a particularly crucial practice for increasing or balancing the recognition classes in a dataset for CNNs [40–42].

## 3. Research Methodology

When approaching a machine learning challenge, the first step is to identify a dataset that accurately defines the problem and train a classifier using it, such as a decision tree, a neural network, or a support vector machine. Manually constructing a dataset is a time consuming, sometimes expensive, and even a risky task. Consider the following scenarios: you want to train a neural network that recognises men overboard, or you want to train a neural network that recognises animals that pass along a road at night. These are specific scenarios that could present many challenges for a researcher, as well as significant cost and time benefit from using graphic modelling software. The pipeline that we suggest for approaching these difficulties is depicted in Figure 1, which begins with the development of a synthetic dataset before moving on to the actual one.

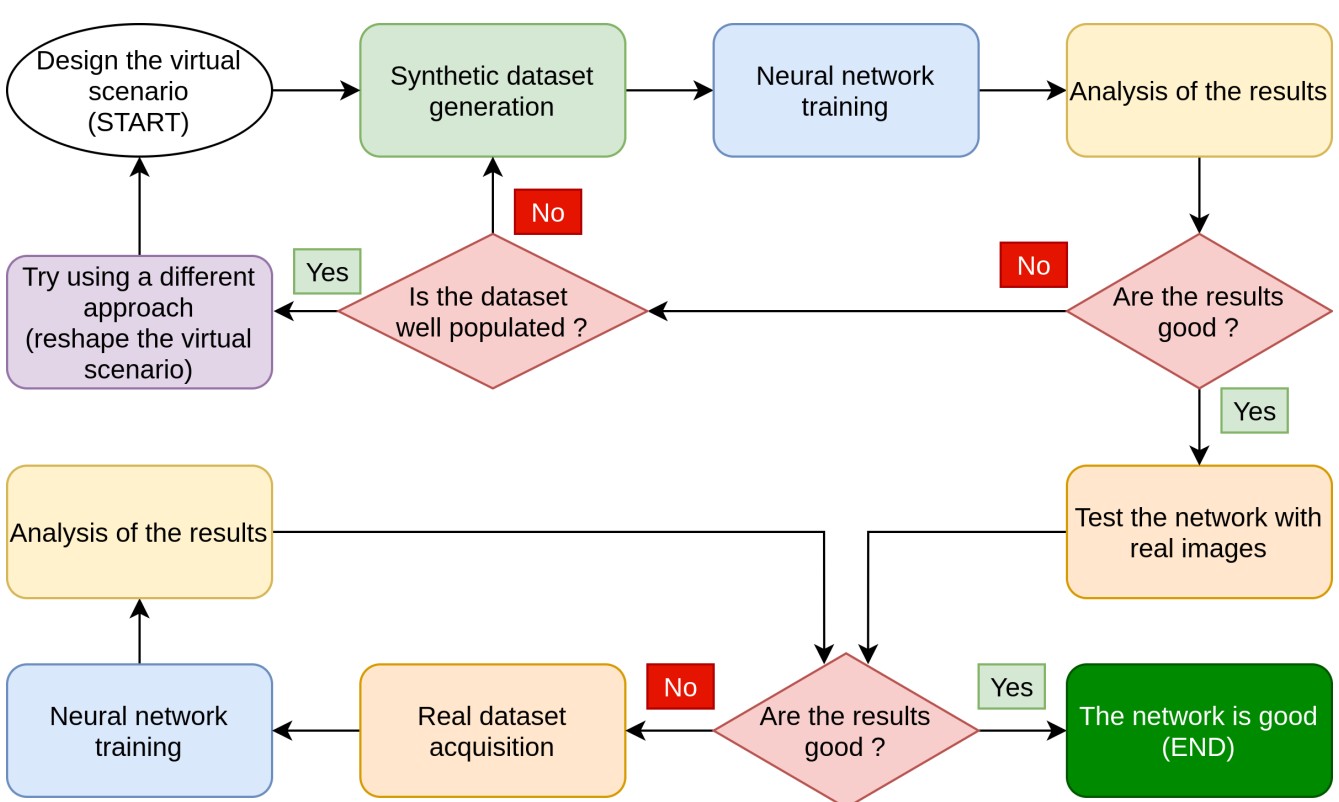

**Figure 1.** Flow chart summarising the various phases of the training of a neural network using virtual scenarios.

### 3.1. Experimental Protocol

The construction of our datasets (each related to the case study dealt with) was carried out by acquiring images once the virtual scenario was created and a script was developed that generates random situations in both cases. In each photograph, the lighting, the shadows, the orientation of the object concerning the camera, and the appearance of random objects (in the case of the men at sea) were varied. In this way, it was possible to generate a very high number of images necessary for learning the neural networks that are described in Section 4. About the use case of men at sea, a second dataset was generated, the validation set, with which we tested the ability of neural networks to recognise the presence of men overboard in new images synthetically generated. Regarding the use case of generating a dataset for the recognition of paintings, we used photographs of real paintings. In the construction of the training set and the validation set, the images were multiplied starting from photographs of the paintings and processing them in the Unity environment, varying the lighting, the orientation for the camera and the shadows. The images used for the construction of the test set with which we measured the accuracy of the neural network in the recognition of paintings through images were not subjected to any processing. The results of the experimental analyses are reported in the graphs and confusion matrices. The graphs will show the trend of the neural networks in the training phases, in the abscissas reported time instants (expressed as the number of epochs), while in the ordinates, the percentage of accuracy in image recognition is reported.

### 3.2. Our Proposed Pipeline

The pipeline that we proposed is now being evaluated. First and foremost, the virtual setting must be designed and built using software such as Unity3D and Blender. This is a crucial stage, and having some example images and a clear sense of how the setting will be constructed might help. After finishing creating the scenario, we can start creating the synthetic dataset. Photographs of the virtual world must be taken in this step, using combinations of light, shadows, and items that we deem appropriate for the research. The acquired photographs can then be used to train a neural network, with care taken to divide the images into two sets, the first of which contains 80% of the samples and can be used to compose the training set, and the second of which contains the remaining 20% and can be used to test the neural network with examples not used in the training phase. After training the neural network, the results must be analysed: if the results are poor, the generated dataset must be double checked, and the scenario modelling may need to be adjusted. If the results are satisfactory, we may assert that our problem can be solved using neural networks, and we can spend time and money looking for and photographing objects in the real world. If financial resources allow, it may be able to replace the synthetic images with real ones and evaluate the dataset generated by the neural network using the continuous learning approach. If the results are positive, the task is completed. Otherwise, we have to go back and examine the dataset of real photographs, train the network again, and see if the goals are met.

Instead of immediately beginning with the capture of real-world images, we advocate starting with synthetic photographs created using 3D modelling software utilising our pipeline. If the three-dimensional environment is created with care, realism, and detail fidelity, we will have a clear idea of the performances that we will be able to obtain in the real world quickly and with low initial costs. Only then can we begin the construction phase of a dataset with photographs taken in the real world.

We discovered several advantages using this method. The first benefit is the speed with which we can generate photographs to train the networks because once we have set up the working environment, we are able to generate an almost infinite number of images by simply running our algorithm and letting it generate images with random combinations of lights, shadows, and objects. Another advantage is the cost: developing a synthetic dataset is far less expensive than creating a genuine dataset, which in our example, includes the hiring of a helicopter, actors, and at least one ship. It should also be remembered that by

utilising a synthetic dataset, we are able to determine much more quickly if the technique we are applying is appropriate or not, and, if required, entirely alter our strategy and attack the problem from different perspectives. We believe that a synthetic dataset should not be used in place of a dataset made up of real images because it is currently impossible to faithfully reconstruct all of the graphic facets and decals that make up the natural world, but we do believe that it is a useful tool for researchers working on machine learning problems, particularly those involving image classification.

In Figure 2, it is possible to see some photographs of the scenario we created. In Figure 2a,b we can see two examples of photos in which there are no men overboard. In Figure 2c,d, there are men overboard, generated randomly and at a random point of the scene. In Figure 3 we find 4 examples of paintings, which make up the dataset of objects we have created. In Figure 3a,b are represented paintings of the Impressionist period, while in Figure 3c,d are represented paintings of the Baroque period.

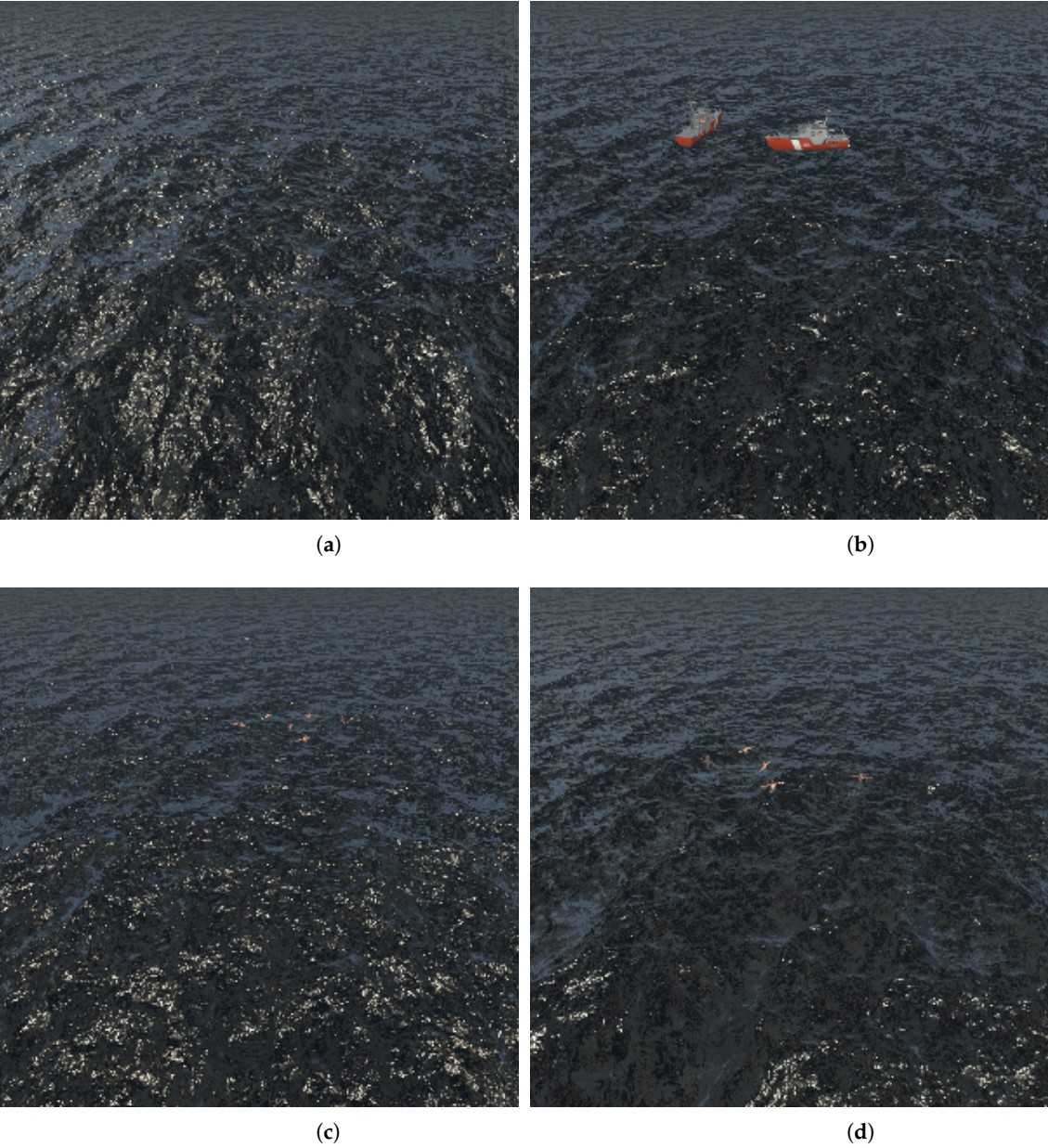

(a)

(b)

(c)

(d)

**Figure 2.** Samples of the men at sea scenarios used in our work. (**a**) Image of the sea without men; (**b**) image of the sea without men, with random objects; (**c**) image of the sea with men; (**d**) image of the sea with men.

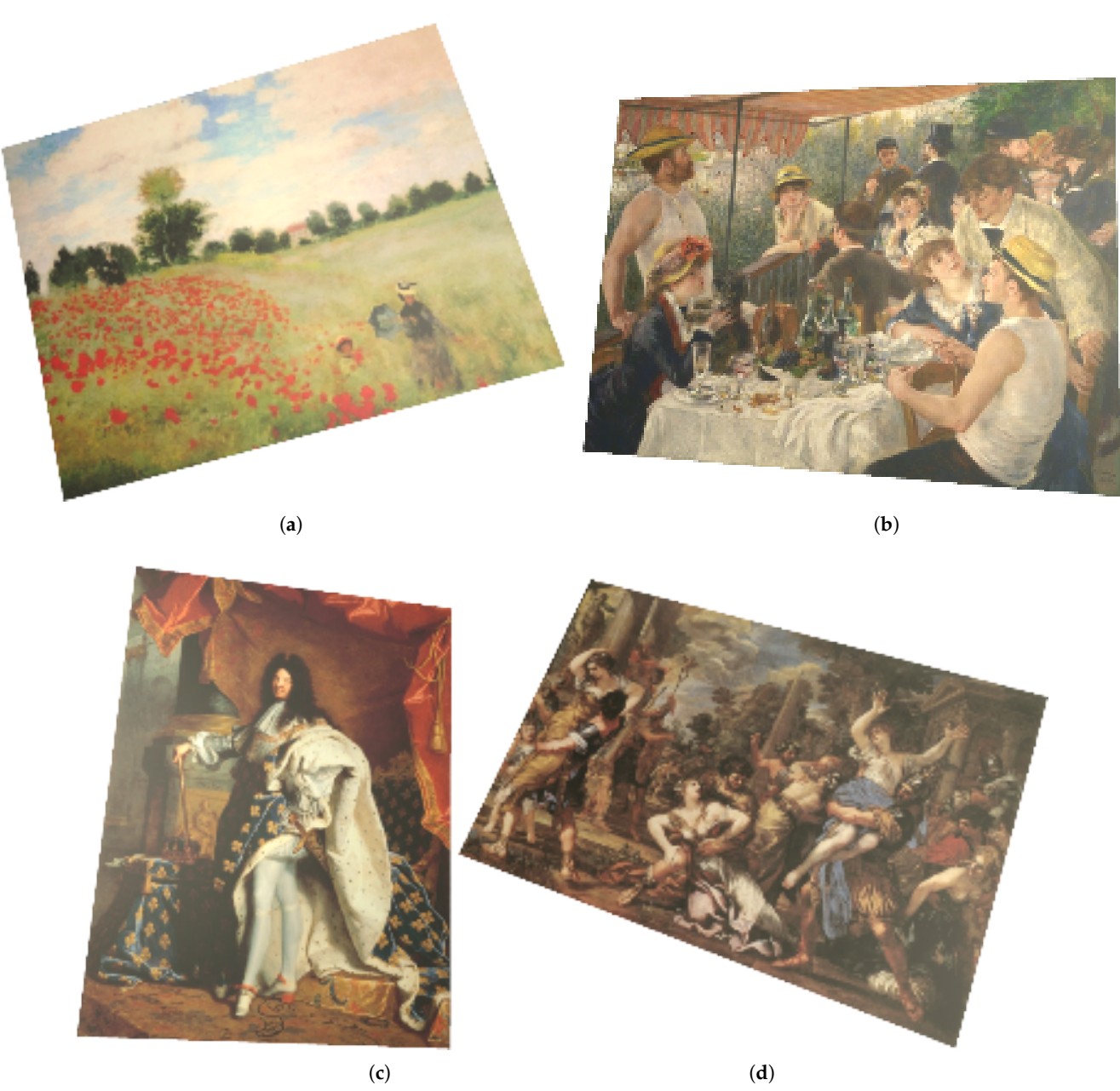

**Figure 3.** Sample paintings used in our work. (**a**) Impressionism; (**b**) Impressionism; (**c**) Baroque;
(**d**) Baroque.

## 4. Discussion of Results

We evaluated the synthetic datasets with neural networks after synthesising them
with virtual reality software, using the methodology outlined in the previous sections. The
goal of the test is to determine whether or not a neural network can operate with pictures
that are not from the real world and whether or not the results it generates are adequate.
The analysis was carried out using two convolutional neural networks. The first network
is Alexnet [43] while the second neural network is InceptionResnet-V2 [44]. We trained
the first neural network via Matlab (https://it.mathworks.com/products/matlab.html
(accessed on 7 October 2021)) software, while the second network was trained via Python
code running on the Google Colab (https://colab.research.google.com/ (accessed on 7
October 2021)) cloud environment. Both networks were trained using the transfer learning
technique [45]. This approach assumed that the networks would be partially trained before

the training began. Before the actual training, the weights of the neural connections were preloaded. The values were derived from the training that these networks performed on the public dataset ImageNet [46]. We then deleted the network's head, as well as the initial 1000-class prediction layer, and added a layer to conduct a binary classification of our photos, for example, by utilising the sigmoid activation function [47]. Finally, after evaluating the general-purpose networks mentioned above, we constructed a customised neural network by modelling the internal structure of the layers to match our proposed problem. This network comprises 12 million neurons, which is a small amount in comparison to general-purpose networks, yet it has produced good performance in the prediction phase on the validation set.

### 4.1. Alexnet

Alexnet was the first neural network we looked at, and we used Matlab software to analyse it. Using the transfer learning method, the neural network was imported and analysed. The network's last three layers were eliminated after it was pre-trained on the ImageNet dataset. These layers acted as learning layers for the network, allowing it to recognise the 1000 ImageNet classes. We replaced these layers with a dense, fully connected layer with a WeightLearnRateFactor of 20 and a BiasLearnRateFactor of 20. A Softmax layer and, lastly, a classification layer were attached to this layer. We examined both the dataset depicting paintings and the dataset representing men at sea with the network setup in this way.

Figure 4 shows the confusion matrices obtained by analysing the dataset of men at sea.

| Training Set | | | | | Validation Set | | | |
|---|---|---|---|---|---|---|---|---|
| TARGET / OUTPUT | False | True | SUM | | TARGET / OUTPUT | False | True | SUM |
| False | 800 / 50.00% | 0 / 0.00% | 800 / 100.00% 0.00% | | False | 179 / 44.75% | 21 / 5.25% | 200 / 89.50% 10.50% |
| True | 0 / 0.00% | 800 / 50.00% | 800 / 100.00% 0.00% | | True | 1 / 0.25% | 199 / 49.75% | 200 / 99.50% 0.50% |
| SUM | 800 / 100.00% 0.00% | 800 / 100.00% 0.00% | 1600 / 1600 / 100.00% 0.00% | | SUM | 180 / 99.44% 0.56% | 220 / 90.45% 9.55% | 378 / 400 / 94.50% 5.50% |
| (**a**) | | | | | (**b**) | | | |

**Figure 4.** Confusion matrices of the sea dataset analysed on Alexnet with Matlab. (**a**) Training set confusion matrix; (**b**) validation set confusion matrix.

Figure 5 shows the confusion matrices obtained by analysing the dataset of the paintings.

| Training Set | | | |
|---|---|---|---|
| TARGET / OUTPUT | Baroque | Impressionism | SUM |
| **Baroque** | 816 / 50.00% | 0 / 0.00% | 816 / 100.00% / 0.00% |
| **Impressionism** | 0 / 0.00% | 816 / 50.00% | 816 / 100.00% / 0.00% |
| **SUM** | 816 / 100.00% / 0.00% | 816 / 100.00% / 0.00% | 1632 / 1632 / 100.00% / 0.00% |

| Validation Set | | | |
|---|---|---|---|
| TARGET / OUTPUT | Baroque | Impressionism | SUM |
| **Baroque** | 204 / 50.00% | 0 / 0.00% | 204 / 100.00% / 0.00% |
| **Impressionism** | 10 / 2.45% | 194 / 47.55% | 204 / 95.10% / 4.90% |
| **SUM** | 214 / 95.33% / 4.67% | 194 / 100.00% / 0.00% | 398 / 408 / 97.55% / 2.45% |

(a)   (b)

**Figure 5.** Confusion matrices of the painting dataset analysed on Alexnet with Matlab. (**a**) Training set confusion matrix; (**b**) validation set confusion matrix.

Figure 6 shows the result of the Alexnet training on the two datasets.

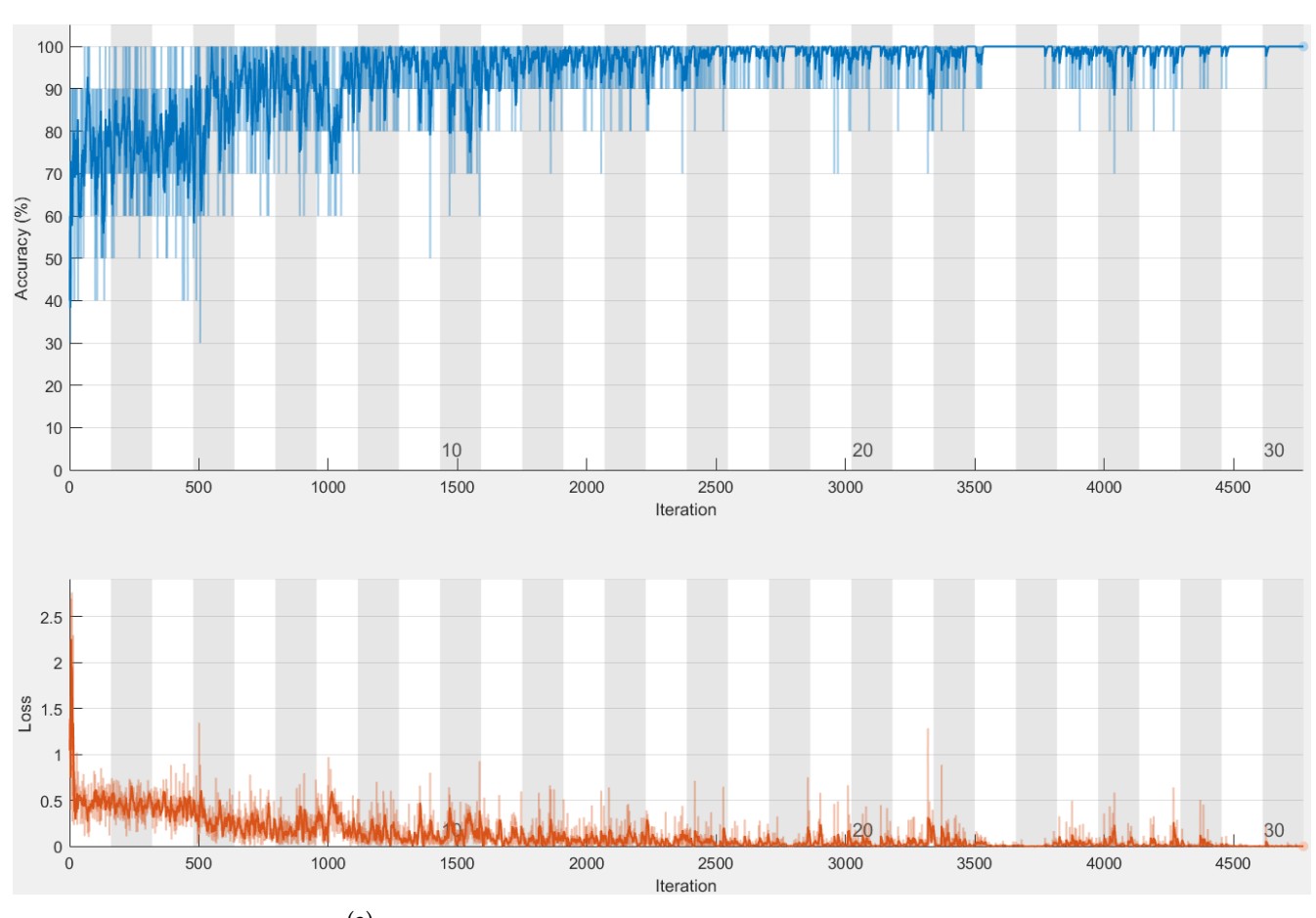

(a)

**Figure 6.** *Cont.*

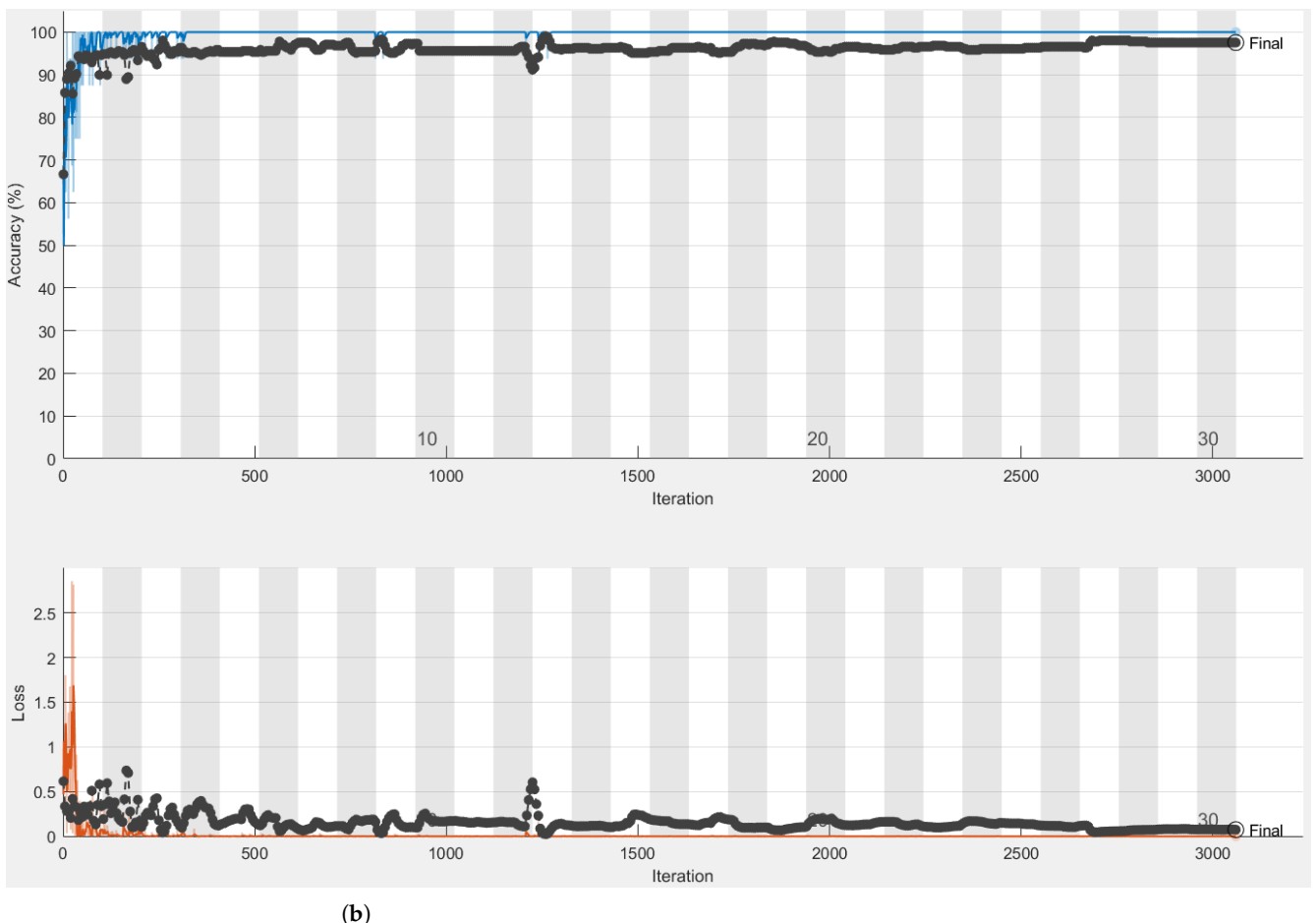

**(b)**

**Figure 6.** Training curve of the datasets analysed on Alexnet with Matlab. (**a**) Training curve of men at sea dataset; (**b**) training curve of paintings dataset.

The graph showing the network's percentage of accuracy as a function of time can be seen in the upper section of the photos. In the lower section of the photos, the loss function as a function of time is presented.

The learning curve of the network in the recognition of the training set is shown in blue in Figure 6a, while the dataset of men at sea is studied. As we can see, the network begins to settle after a few early oscillations, becoming increasingly precise.

Figure 6b, on the other hand, depicts the network's learning curve as it analyses the collection of paintings. Blue highlights the various training iterations. We have added periodic checks in the recognition of the validation set as an extra analysis in this example, and the interpolation of the recognition percentages achieved is shown by the black curve. Figure 4b shows the final result in the recognition of the validation set of the man overboard dataset.

The findings acquired from the first tested network are, in our perspective, extremely good; in particular, we can see how the validation set paintings are identified with an accuracy of 97.5%. The accuracy of the dataset of men at sea, on the other hand, is lower, with an overall accuracy of 94.5%. When in the confusion matrix, we consider the cell corresponding to the column "true" and the row "true", that is, when there were men overboard, we can see that 199 of the 200 photos in the sample were properly identified. The Alexnet neural network described above takes up a total of 201.9 MB on disk.

### 4.2. InceptionResNet-V2

We used the Google Colab cloud environment to test our second system, the InceptionResNet-V2 neural network. This network was also preloaded using the transfer

learning approach; thus, the weights of the neurons were already capable of classifying the 1000 ImageNet classes. We imported the network, removed the network's head, which was made up of dense learning layers, and replaced it with the following layers. The first is a flatten type layer, whose job was to convert the network's data structure into a float vector. After that, we added two dense layers of 64 neurons with rectified linear unit (ReLU) activation functions to learn the features collected from the network's convolutional layers. Finally, we added a layer that just contains one neuron and is responsible for binary categorization. Binary cross entropy (https://keras.io/api/losses/probabilistic_losses/ (accessed on 7 October 2021)) was employed as the loss function, while Adam (https://keras.io/api/optimizers/adam/ (accessed on 7 October 2021)) [48] was chosen as the optimizer. In a cloud environment like Google Colab, the network described above has a total of 60,632,481 parameters, which is a big quantity to maintain and handle. As a consequence, we devised a strategy that enabled us to operate effectively with such a large network. The early layers of the InceptionResNet-V2 network were frozen, and only the final layers were permitted to be trained. The layers that extract picture characteristics and are pre-trained on ImageNet were not impacted in this way. Because of this approach, 54,336,736 parameters are immutable and do not change during the network's training, whereas the remaining 6,295,745 parameters need to be trained to learn how to accurately categorise our datasets. We have additionally set up two more aspects. The first is in the training phase: we assessed the network accuracy percentage on the validation set every time an epoch finished. If the accuracy rate increased, the network model was saved; this step is considered a checkpoint. In this method, even if the network overfits during training and its accuracy on the validation set deteriorates, we may still trace the optimum combination of parameters gained during training. The second point to consider is when to halt (*early stopping*). We can track the development of the network training phase and verify its accuracy in identifying the dataset using this method. After each period, a check was performed. If the network does not increase recognition accuracy for a predefined number of epochs (generally three), we may say that we have found a local minimum in the range of possible solutions to the problem. This abnormality is detected, and network training is halted, saving time that would otherwise be squandered.

Figure 7 shows the confusion matrices obtained by analysing the dataset of men at sea with the InceptionResNet-V2 neural network. In this case, the network recognises each image provided through the validation set, reaching an accuracy of 100%. The confusion matrices are derived by examining the dataset of the paintings shown in Figure 8; the validation set has a very high accuracy percentage of 98.03%.

| Training Set | | | |
|---|---|---|---|
| TARGET ⟍ OUTPUT | False | True | SUM |
| False | 799<br>49.94% | 1<br>0.06% | 800<br>99.88%<br>0.12% |
| True | 0<br>0.00% | 800<br>50.00% | 800<br>100.00%<br>0.00% |
| SUM | 799<br>100.00%<br>0.00% | 801<br>99.88%<br>0.12% | 1599 / 1600<br>99.94%<br>0.06% |

(**a**)

| Validation Set | | | |
|---|---|---|---|
| TARGET ⟍ OUTPUT | False | True | SUM |
| False | 201<br>50.25% | 0<br>0.00% | 201<br>100.00%<br>0.00% |
| True | 0<br>0.00% | 199<br>49.75% | 199<br>100.00%<br>0.00% |
| SUM | 201<br>100.00%<br>0.00% | 199<br>100.00%<br>0.00% | 400 / 400<br>100.00%<br>0.00% |

(**b**)

**Figure 7.** Confusion matrices of the men at sea dataset analysed on InceptionResNet-V2 with Google Colab. (**a**) Training set confusion matrix; (**b**) validation set confusion matrix.

| Training Set | | | |
|---|---|---|---|
| TARGET<br><br>OUTPUT | Baroque | Impressionism | SUM |
| **Baroque** | 778<br>47.67% | 39<br>2.39% | 817<br>95.23%<br>4.77% |
| **Impressionism** | 0<br>0.00% | 815<br>49.94% | 815<br>100.00%<br>0.00% |
| **SUM** | 778<br>100.00%<br>0.00% | 854<br>95.43%<br>4.57% | 1593 / 1632<br>97.61%<br>2.39% |

(**a**)

| Validation Set | | | |
|---|---|---|---|
| TARGET<br><br>OUTPUT | Baroque | Impressionism | SUM |
| **Baroque** | 195<br>47.79% | 8<br>1.96% | 203<br>96.06%<br>3.94% |
| **Impressionism** | 0<br>0.00% | 205<br>50.25% | 205<br>100.00%<br>0.00% |
| **SUM** | 195<br>100.00%<br>0.00% | 213<br>96.24%<br>3.76% | 400 / 408<br>98.04%<br>1.96% |

(**b**)

**Figure 8.** Confusion matrices of the painting dataset analysed on InceptionResNet-V2 with Google Colab. (**a**) Training set confusion matrix; (**b**) validation set confusion matrix.

Furthermore, we report in Figure 9 the graphs obtained with the training of the neural network.

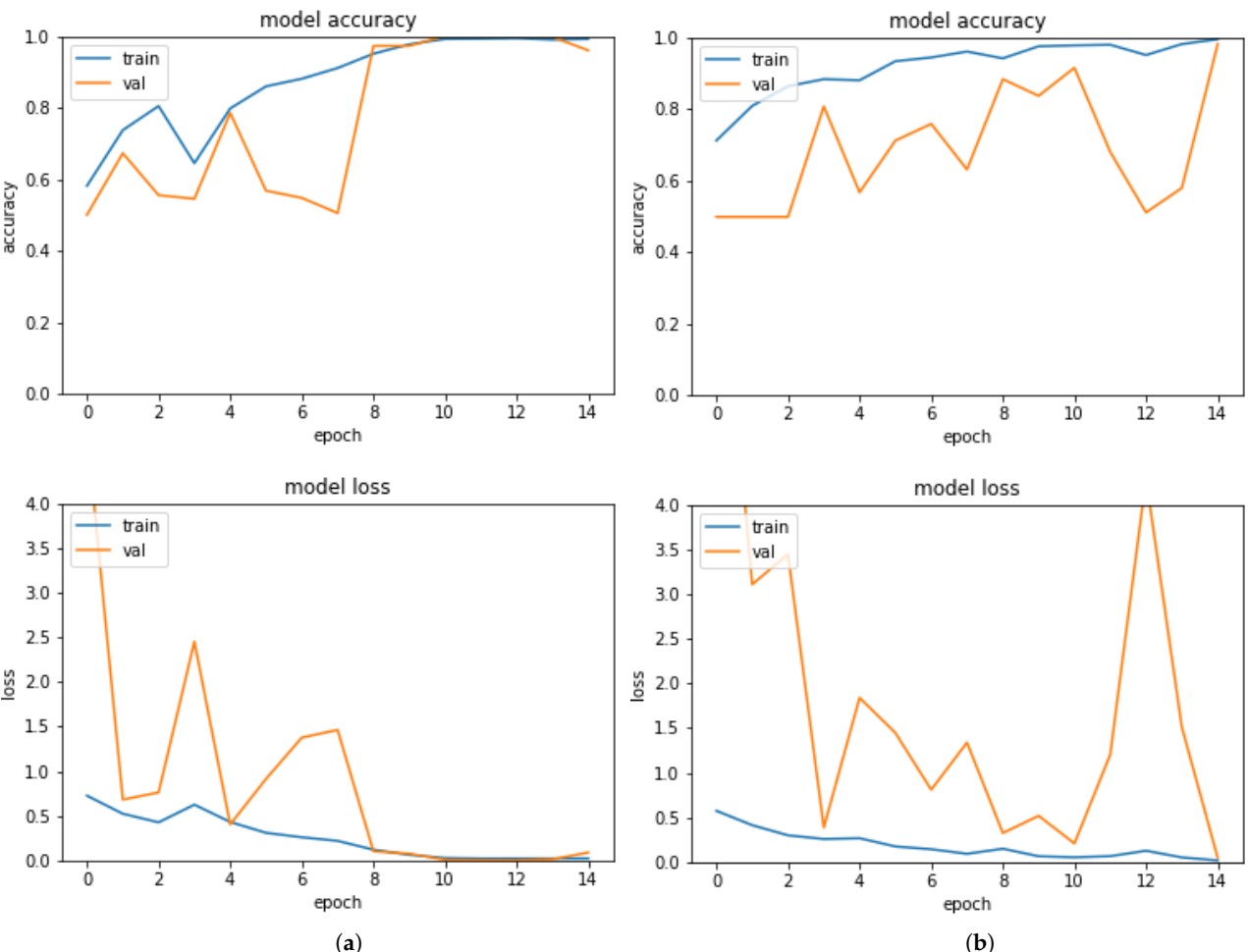

(**a**)

(**b**)

**Figure 9.** Training curve of datasets analysed on InceptionResNet-V2 with Google Colab. (**a**) Sea dataset; (**b**) paintings dataset.

The graphs show how the proportion of accuracy has increased over time, peaking at epoch n.10 and then declining. The network then entered an overfitting phase, during which the accuracy of the validation set decreased significantly. We were able to save the condition in which the network was at its highest level of accuracy, which was obtained around epoch n.10, thanks to early stopping and checkpoints. The neural network as described took up 54.5 MB of storage space. The convolutional layers were frozen; thus, no changes in the weight values of the neural interconnections were made compared to the network trained on ImageNet, allowing for such a small amount of space to be filled.

### 4.3. Custom Convolutional Neural Network

To finish our study, we constructed a custom neural network for the collection of paintings. The goal of this network's architecture is to achieve a more linear training phase than InceptionResNet-V2. Indeed, we want to avoid the frequent spikes and decays on accuracy that we saw with the general-purpose network, and we want the loss function to be smoother. In this approach, we want to develop a neural network that is theoretically more stable when applied to pictures that are not synthetically created. Figure 10 depicts the network we are presenting.

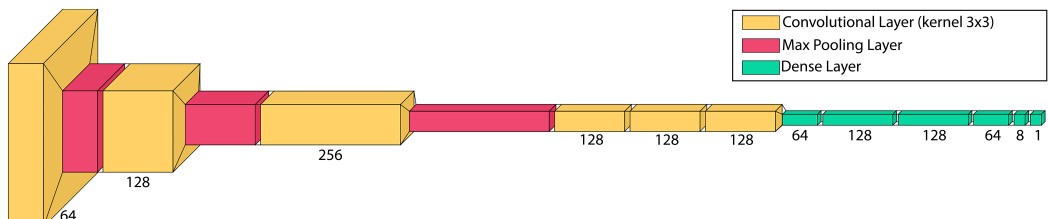

**Figure 10.** Structure of the custom convolutional neural network.

There are three blocks to its construction. The identification of the highest level characteristics is the initial block. We employ a convolutional layer sequence and max-pooling to do this. The three first convolutional layers have 64, 128 and 256 filters, respectively, as we go deeper into the network. The second block is made up of three convolutional layers, each with 128 filters. These layers are responsible for learning the finer details of our datasets. A series of dense layers with ReLU activation functions compose the third block. These layers are responsible for learning and categorising the characteristics retrieved by convolutional layers. The network's last layer is composed of a single neuron with a sigmoid activation function. Adam was the optimizer for this neural network, and there were a total of 12,209,553 parameters to train. The setup of the neuronal weights of this neural network required 139.8 MB of storage space. Figure 11 shows the confusion matrices created by testing the neural network on the datasets. As it can be seen, the recognition percentages are quite high, with a validation set accuracy of 97.54%.

Figure 12a,b show the statistics gained during the training phase, whereas Figure 12d shows the Roc curve. As we may see, the learning curve is highly steady, and the validation set's identification percentage is very close to the training set's recognition accuracy. Checkpoints and early stopping were also implemented in this situation, allowing us to save the optimal configuration of neural weights that the training phase could create.

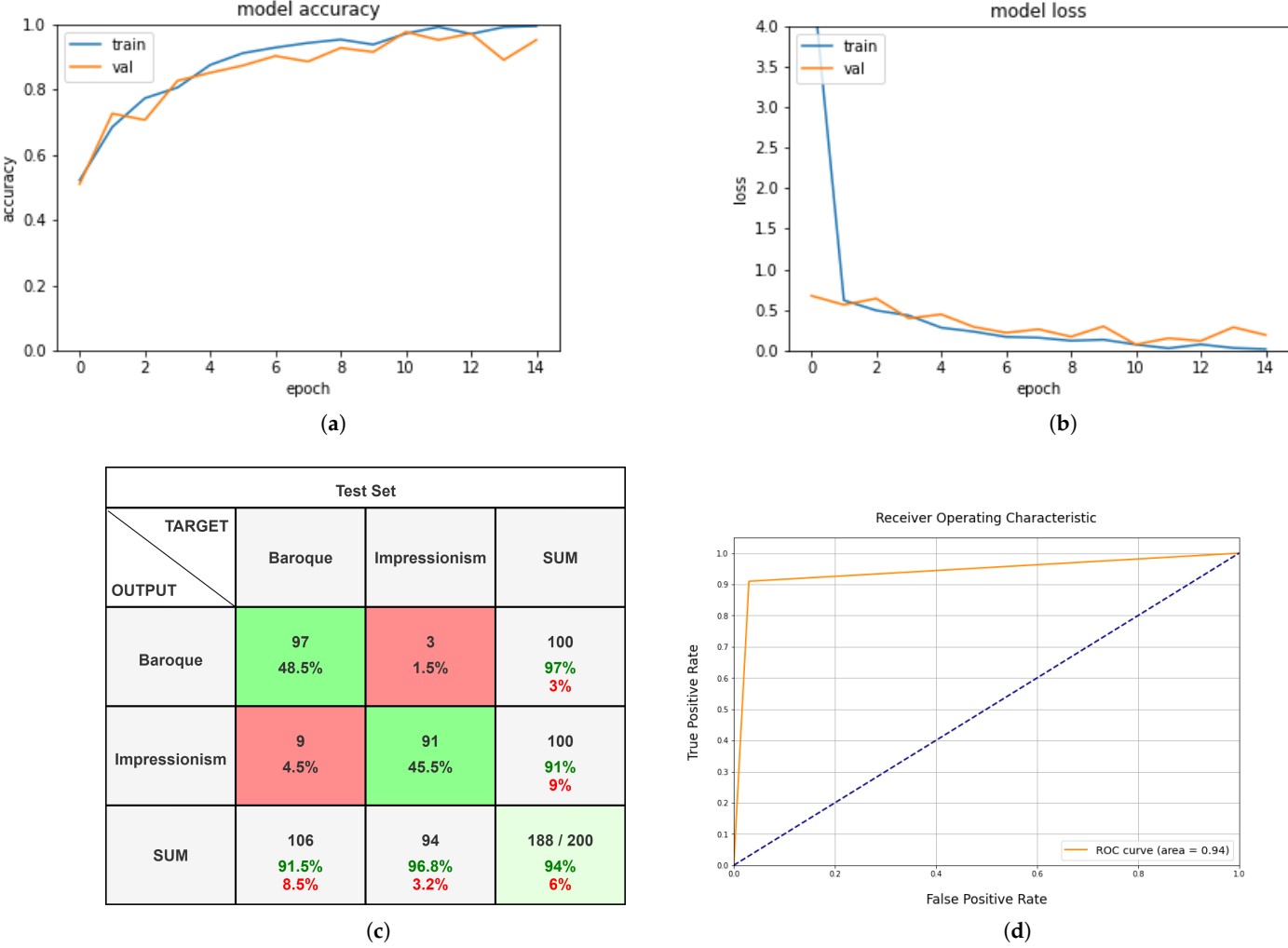

**Training Set**

| TARGET OUTPUT | Baroque | Impressionism | SUM |
|---|---|---|---|
| Baroque | 804 / 49.26% | 4 / 0.25% | 808 / 99.50% 0.50% |
| Impressionism | 2 / 0.12% | 822 / 50.37% | 824 / 99.76% 0.24% |
| SUM | 806 / 99.75% 0.25% | 826 / 99.52% 0.48% | 1626 / 1632 / 99.63% 0.37% |

(**a**)

**Validation Set**

| TARGET OUTPUT | Baroque | Impressionism | SUM |
|---|---|---|---|
| Baroque | 206 / 50.49% | 6 / 1.47% | 212 / 97.17% 2.83% |
| Impressionism | 4 / 0.98% | 192 / 47.06% | 196 / 97.96% 2.04% |
| SUM | 210 / 98.10% 1.90% | 198 / 96.97% 3.03% | 398 / 408 / 97.55% 2.45% |

(**b**)

**Figure 11.** Confusion matrices of the paintings dataset analysed on Custom-CNN with Google Colab. (**a**) Training set confusion matrix; (**b**) validation set confusion matrix.

**Test Set**

| TARGET OUTPUT | Baroque | Impressionism | SUM |
|---|---|---|---|
| Baroque | 97 / 48.5% | 3 / 1.5% | 100 / 97% 3% |
| Impressionism | 9 / 4.5% | 91 / 45.5% | 100 / 91% 9% |
| SUM | 106 / 91.5% 8.5% | 94 / 96.8% 3.2% | 188 / 200 / 94% 6% |

(**c**)

(**d**)

**Figure 12.** Confusion matrix and statistical analysis of the paintings dataset analysed on Custom-CNN with Google Colab. (**a**) Accuracy function; (**b**) loss function; (**c**) confusion matrix paintings test set; (**d**) receiver operating characteristic curve.

The best configuration was found at the end of epoch n.11; however, the early stopping of the training at epoch n.14, as the network was approaching overfitting, caused the training to be terminated. Finally, we conducted a deeper investigation. We created a test set of paintings that were not included in our synthetic dataset and utilised it to see if the neural network we trained could distinguish the painting style. The analysis gave very good results with an accuracy percentage of 94%. The confusion matrix constructed on the test set as presented is shown in Figure 12c.

Figure 13 shows the table comparing the results on the recognition of the men at sea dataset, according to the chosen metrics of the two neural networks tested, Alexnet and InceptionResNet-V2. It can be seen that for validation accuracy, the InceptionResNet-V2 network has a better validation accuracy. Figure 14 shows the table comparing the results on the recognition of the paintings dataset, according to the chosen metrics, of the three neural networks tested, Alexnet, InceptionResNet-V2 and the custom network. It is noted that for the validation accuracy, the InceptionResNet-V2 network shows a better validation accuracy, but the custom network has several trainable parameters equal to 1/5 compared to Inception.

|  | Alexnet | InceptionResnet-V2 |
|---|---|---|
| Accuracy | 100.00% | 99.63% |
| Validation accuracy | 94.50% | 100.00% |

**Figure 13.** Comparing between Alexnet and InceptionResNetV2 CNNs for the use case of the men at sea.

|  | Alexnet | InceptionResnet-V2 | Custom |
|---|---|---|---|
| Accuracy | 100.00% | 97.61% | 99.63% |
| Validation accuracy | 97.54% | 98.03% | 97.54% |

**Figure 14.** Comparing among Alexnet, InceptionResNetV2 and the Custom CNNs for the use case of the paintings.

## 5. Best Practices Generating a Synthetic Dataset in Virtual Environments

This section describes the techniques we recommend for generating synthetic datasets. In Section 5.1, we describe the techniques we recommend for generating synthetic scenarios using Unity3D, while in Section 5.2 we describe how to generate synthetic datasets depicting objects. The techniques described can be applied generically to many other graphic modelling software; we used Unity3D only as a use case.

### 5.1. Representation of the Synthetic Scenario

The first step is to use graphic modelling software to properly represent the scenario. One of the most well-known and widely used programmes to do this is Blender. This software product also has a significant community that has created a lot of tutorials that may be helpful to individuals who are just getting started in the realm of graphic modelling. After receiving the model, it is necessary to import it into Unity3D. It is sufficient to place the model that we produced into the gameobject after constructing an empty gameobject to use as a container. As a result, a camera must be placed in the scene and positioned so that it frames precisely the scene, as illustrated in Figure 15.

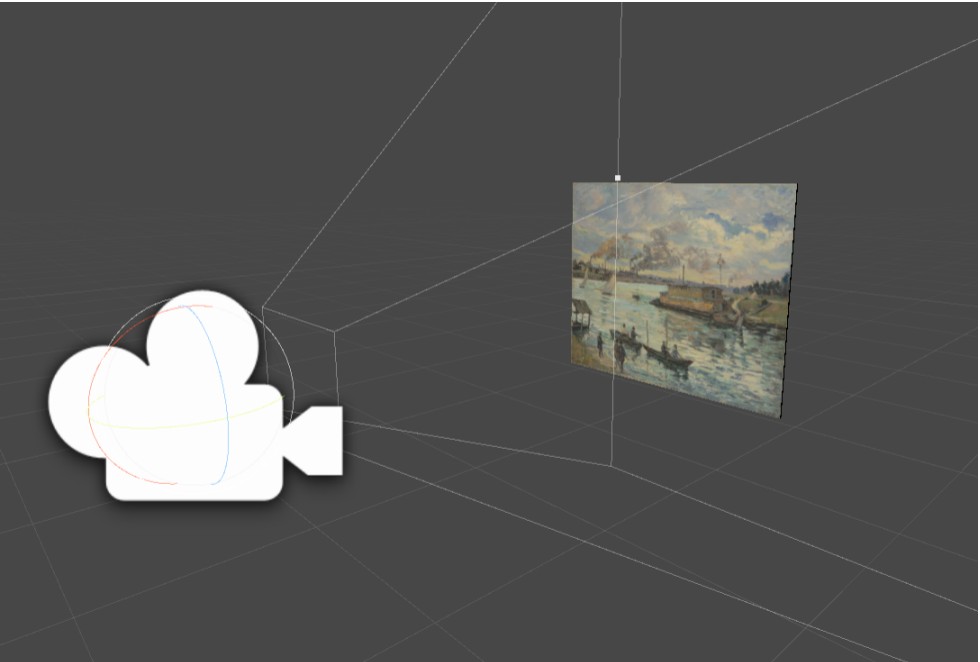

**Figure 15.** Camera that precisely frames the scene.

Depending on the lighting conditions we want to reproduce, we also need a directional light and maybe one or more pointlights or spotlight types of lights. After completing these preparatory procedures, we need to write a script that changes the lighting conditions at random, rotate the item, and take photos with the appropriate resolution. We propose including some public variables in the script to define the parameters using the Unity3D graphical interface (the Inspector) rather than having to change the code all the time. Some examples of public variables are the path to the output image folder, the number of photographs to be taken, the object and light's minimum and maximum rotations, the light's minimum and maximum intensities, and so on. Figure 16a shows an example.

The operations flow that we suggest is as follows. Execute the following procedures within a loop that iterates a number of times equal to the number of photos we wish to generate:

1.  Random rotation of the object: we produce three random integers that are contained in the minimum and maximum rotation ranges that we previously established and assign them as coefficients of the object's X, Y, and Z rotations.
2.  Changing the scene's global lighting: We produce three random integers that represent the potential rotations of the light in the scene. It is critical to establish the ranges of the three variables accurately to guarantee that the representation obtained is believable. A picture with illumination from the bottom up, or even directly into the viewer's eyes, for example, would be unusual. As a result, we recommend trying until the appropriate outcome is obtained. These numbers are then allocated as light rotation coefficients once they are formed. It is also possible to produce a random value that alters the light's intensity and hue.
3.  Acquisition of image: to store a photograph of the scene, generate a rectangle that overlaps the user interface starting at the $x_0, y_0$ coordinates of $(0, 0)$ and finishing at the $x, y$ coordinates of (Screen.width, Screen.height), then extract the RGB values of the pixels included inside the rectangle.
4.  Scaling the image: after we have gotten the RGB values, we need to scale the image to fit the size requirements of the neural network we are going to utilise. For example, if we are creating a dataset to train the InceptionResNetV2 network, we may scale the photos directly in Unity3D to $299 \times 299$ resolution.

5. Saving the image: once the pixels are captured and resized, the image must be saved to a file system, for example, in the PNG format. During the saving step, we propose distinguishing the objects using an identifying name and an incremental integer that is used to create the saved file's name, such as `"impressionism_0000X.png"`.

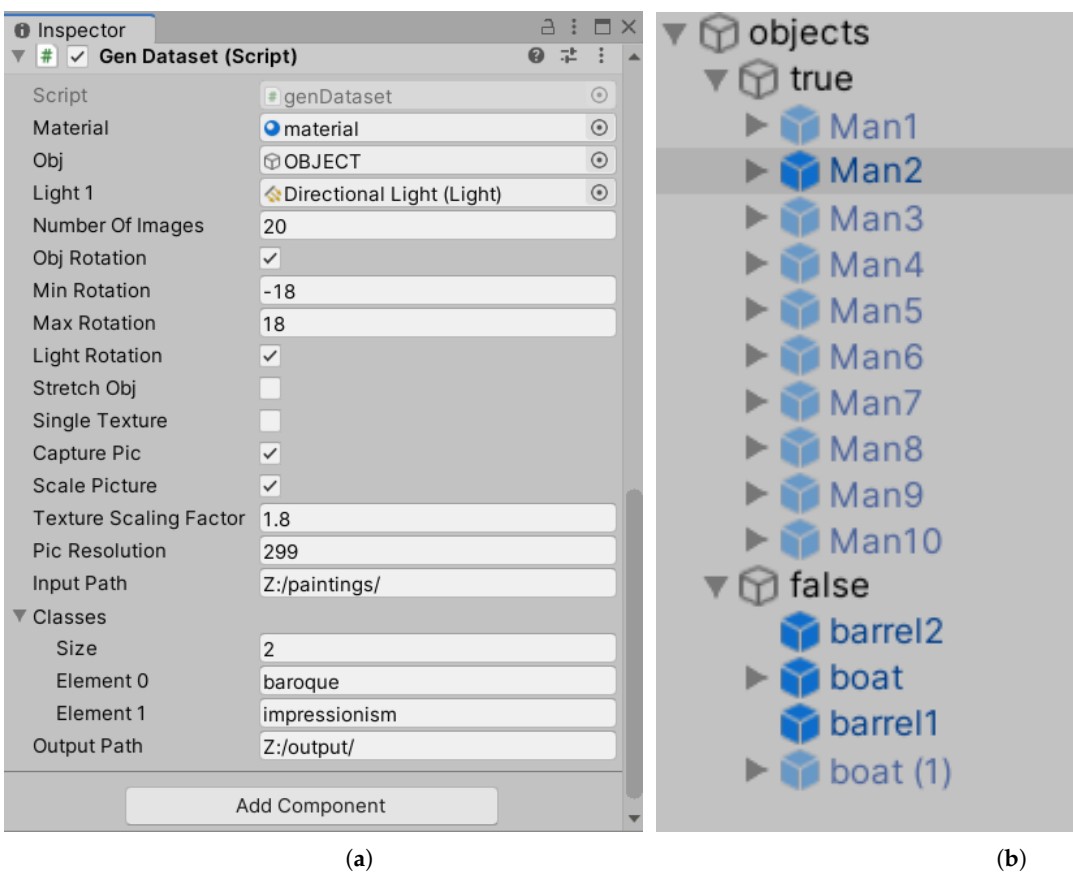

(**a**)                      (**b**)

**Figure 16.** Unity3D interface. (**a**) Inspector, the script and the parameters; (**b**) Objects in the sea.

*5.2. Dataset for the Classification of Images Representing Objects*

The following are best practices for generating virtual scenario datasets using Unity3D. Now let us assume we want to do a true-false binary classification. First and foremost, the models of the things that make up the scene must be created. The models must be realistic, and we suggest Blender in this situation as well. It is also important to consider how to construct the examples for the *true* and *false* classes. We must also ensure that the items are not always arranged in the same area of the image; to do so, we need to designate locations where they can be formed and then displayed to the camera. We created a scenario suitable for both photographs associated with the *true* category and photographs associated with the *false* category in which we represented men at sea, shipwrecked people, and created a scenario suitable for both photographs associated with the *true* category and photographs associated with the *false* category. The images in the *true* category account for half of the whole dataset. The remaining 50% are in the *false* category, which is split into two sub-categories, with half of them representing simply the scenario with the sea and random light conditions. The other half is a representation of the sea, with ships and barrels floating on the surface. First of all, we recommend generating and putting a container object, in our case "*objects*", into the scene. We added two more containers to it, one for objects of the *true* class and the other for objects of the *false* class. Three-dimensional models of people, ships, and barrels were placed within the two containers. Figure 16b provides an example.

As a result, the flow of the operations that we suggest is as follows. The following procedures should be executed within a loop that iterates a number of times equal to the number of images we wish to generate:

1. Restore the scene to its original state by hiding all items on the scene, except the sea, the camera, and the sunlight at the start of each new iteration.
2. Make a random number of the objects active (and so display them within the scene) while producing pictures of the class *true/false*.
3. Alter the location of the objects: for each object in the scene, we generate three random numbers (X, Y, and Z) from the range of coordinates that the camera can frame and update the position of this object to the produced coordinates.
4. Change the rotation of objects: for each object in the scene, we produce three random integers (X, Y, and Z) that are within the desired rotation interval and realistically match the class to be formed. Then, using the provided values, rotate the item under examination.
5. Modify the lighting of the scene as described in point 2 in the previous list.
6. Acquire the scene image as specified in the preceding list's point 3.
7. Resize the image to the size acceptable by the neural network we wish to test, as explained in the preceding list's point 4.
8. Save the picture to the file system as stated in point 5 from the preceding list, be sure to give each image a name that allows you to identify the class to which it belongs.

## 6. Conclusions and Future Works

We demonstrated that synthetic datasets may be a valuable resource for researchers utilising machine learning algorithms to identify objects or scenarios. When dealing with challenges of this nature, you often have two options: either use datasets that other scientists have made accessible on the internet, or invest time and money in constructing an ad hoc dataset specific to the topic at hand. It takes time and money to collect materials and images to create a dataset, which generally comprises tens of thousands of shots. Taking the photos required to train a neural network might be risky in some circumstances. We can naturally think, for example, of the training of the network of men at sea, the recognition of animals or pedestrians along the motorways for the automatic braking systems of cars, and others. The synthetic datasets created by the pipeline and the methodologies presented in this article allow us to accelerate this process and predict which kinds of images will perform better for the task at hand. We also believe that, as a result of the high degrees of realism achieved by computer graphics, the image quality is quite good and will continue to improve, allowing the construction of increasingly realistic datasets. We are going to expand our study in the future and concentrate on particular elements, such as the union of synthetic and realistic datasets, by examining how neural networks trained on synthetic datasets react while adding instances to the original dataset and utilising continuous learning approaches.

**Supplementary Materials:** The open source code for the generation of the confusion matrices developed for this paper is available on GitHub https://github.com/DamianoP/confusionMatrixGenerator (accessed on 15 November 2021).

**Author Contributions:** Conceptualization, D.P., M.S. and O.G.; data curation, D.P., M.S. and O.G.; investigation, D.P., M.S. and O.G.; methodology, D.P., M.S. and O.G.; software, D.P. and M.S.; supervision, O.G.; validation, D.P. and M.S.; writing—original draft, D.P., M.S. and O.G.; writing—review and editing, D.P., M.S. and O.G. All authors have read and agreed to the published version of the manuscript.

**Funding:** This research received no external funding.

**Institutional Review Board Statement:** Not applicable.

**Informed Consent Statement:** Not applicable.

**Acknowledgments:** We thank Google for making the Google Colab cloud environment available to researchers. We thank Matlab for the opportunity given to use their development environment.

**Conflicts of Interest:** The authors declare no conflict of interest.

**Abbreviations**

The following abbreviations are used in this manuscript:

| | |
|---|---|
| CNN | Convolutional Neural Network |
| GAN | Generative Adversarial Network |
| RAN | Recurrent Adversarial Network |
| UV | The u,v graphic coordinates |
| VR | Virtual Reality |
| ROC | Receiver Operating Characteristic |
| NN | Neural Network |

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
