# Peer review of "Synthetic Data Generation to Speed-Up the Object Recognition Pipeline"

_electronics, doi:10.3390/electronics11010002_

Round 1

Reviewer 1 Report

  1. The whole manuscript requires thorough proofreading and grammatically corrections.
  2. Line 36-42, the text is divided into too many paragraphs. A single paragraph describing objective, deliverables and layout of the manuscript should be added instead of many small paragraphs.
  3. There is no reference of sea (Fig. 2) in the text. Also, the caption of Fig.2 should be more descriptive, illustrating the purpose of the figure.
  4. The position of Fig. 4 and Fig.5 should be switched as men at sea should come before the paintings.
  5. 6 should be re plotted with correct font size of x and y labels.
  6. The position of Fig. 7 and Fig.8 should be switched as men at sea should come before the paintings.
  7. Is there a reason why two different software (MATLAB and Python) were used? Evaluating networks using one software would make the comparison more convincing.
  8. A table summarizing the comparison between Alexnet, InceptionResNet and Custom convolutional neural network should be added.
  9. Only two scenarios (men at sea and painting) were used to train and test neural networks. Addition of more scenarios will make the evaluation and comparison more conclusive and robust.

Author Response

Response to Reviewer 1

Dear reviewer,
  all the authors would like to thank you for your positive comments on our work and for your suggestions, which have enabled us to greatly improve the quality of our work.
We describe below the actions taken in correspondence with your valuable indications:

  1. Done
  2. The named paragraphs have been revised according to your suggestions 
  3. We included the missing reference, 
  4. The figures have been swapped. 
  5. The two figures have been enlarged by placing them one on top of the other instead of side by side, in order to make better use of the printing area and thus make them easier to read.
  6. The figures have been swapped. 
  7. The choice of using Matlab on the one hand and Google Colab on the other stems from the fact that the Alexnet network is preloaded in Matlab, while InceptionResnetv2 is available in Google Colab. In the same environment we implemented the custom neural network. The results we have shown focus on the performance of the different neural networks.
  8. We added the comparison among the three neural networks related to the two selected scenarios. Thank you so much for suggesting this important improvement. 
  9. The topic of our work is very innovative and we hope that its importance will be grasped so that our methodology can be incorporated into the implementation of neural networks by our colleagues. In fact, our approach makes it easy to significantly increase the amount of data required for the correct preparation of any neural network. For this reason we have selected, among the many situations that we could have chosen, two scenarios that involve a very diversified use of the virtual reality framework. The first aims at the creation of synthetic scenarios and the second at the generation of a set of images that starting from a subset of images or objects determines a much larger set (by modifying lighting, orientation, etc.), necessary for the formation of the neural network.

Reviewer 2 Report

  1. How to evaluate the fidelity of the generated images?
  2. How to enhance the effeciency of the generating process? For example, the advantage and difference in comparison with data augmentation in AI. 
  3. Are there some semantic considerations in this technique? The generated content is always reasonable? 

Author Response

Response to Review 2

Dear reviewer.
  all the authors would like to thank you for your positive comments on our work and for your suggestions, which have enabled us to greatly improve the quality of our work.

The topic of our work is very innovative and we hope that its importance will be grasped so that our methodology can be incorporated into the implementation of neural networks by our colleagues. In fact, our approach makes it easy to significantly increase the amount of data required for the correct preparation of any neural network. For this reason we have selected, among the many situations that we could have chosen, two scenarios that involve a very diversified use of the virtual reality framework. The first aims at the creation of synthetic scenarios and the second at the generation of a set of images that starting from a subset of images or objects determines a much larger set (by modifying lighting, orientation, etc.), necessary for the formation of the neural network.

Considering your valuable indications, we can say that:

  1. The evaluation of the images generated in the first use case where synthetic scenario generation is used can only be qualitative at this stage. A performance test of the live neural network (e.g. installed on a drone inspecting portions of the Mediterranean Sea) would be the best quantitative measure of the goodness of the virtual scenario generated. In the other use case, we start from existing images that are then reprocessed without loss of quality, if not desired for a better training of the neural network.
  2. The first use case, relating to men at sea, describes a synthetic situation and produces data in an absolutely innovative way by creating a virtual scenario which, thanks to the performance of the hardware and the graphic frameworks, can be extremely realistic.
    The second use case, concerning paintings, is exactly an evolution of the classic AI data augmentation process. In fact, in this case we are able to manipulate the original image not only by rotating the subject but also by varying light, shadows and orientation with respect to the observer.
  3. The first use case, concerning men at sea, requires extreme care and attention in its realisation. Obviously, serious consideration must be given to the semantics and significance of the scene, which inevitably determine the degree of truthfulness.
    Also in the second use case, relative to the paintings, it is necessary that plausible conditions and parameters of illumination and of framing are chosen in order to create a set of significant data for the training of the neural network.

Reviewer 3 Report

The authors present a methodology that will generate artificial image datasets used for training neural networks. 

The state of art is well written and correctly presents the case for artificial image dataset generation.

The methodology is interesting but some questions can be raised:

  • The experimental results could eventually validated against a real dataset;
  • figure 12. d) shows the ROC and it is strange, is it correct?
  • in the paper figure 12d is referred as 9d?

The experimental protocol is poorly defined. The set of measurements should also include some measure of error like de number of significative digits or standard deviation associated to the results. 

Minor corrections of English are advisable. For example, 

  • line 89 "practise" should be "practice";
  • line 54 "The section 3" should be "Section 3";
  • line 34 "number of" should be removed;
  • line 151 "python code" should be "Python code". 

and many more situations exist. Nevertheless the English is mostly good.

Author Response

Response to Review 3

Dear reviewer,
  all the authors would like to thank you for your positive comments on our work and for your suggestions, which have enabled us to greatly improve the quality of our work.
The topic of our work is very innovative and we hope that its importance will be grasped so that our methodology can be incorporated into the implementation of neural networks by our colleagues. In fact, our approach makes it easy to significantly increase the amount of data required for the correct preparation of any neural network. For this reason we have selected, among the many situations that we could have chosen, two scenarios that involve a very diversified use of the virtual reality framework. The first aims at the creation of synthetic scenarios and the second at the generation of a set of images that starting from a subset of images or objects determines a much larger set (by modifying lighting, orientation, etc.), necessary for the formation of the neural network.

The changes we have made, which reflect your suggestions, are highlighted in red.
Some words have remained unchanged as we have used British English.
As per the observation of the experimental protocol we have created a new subsection (“The experimental protocol”) detailing it.
As far as the model error is concerned, it is explained by the trend of the loss curve, which expresses the variation of the error during the learning phase of the model in relation to the epochs.

As per the note about the ROC figure, we can say that the trend of the ROC curve that we found is related to the accuracy and precision of our model, which are very high. Similar trends are easily seen in literatures (e.g.

https://www.researchgate.net/publication/329807137_Predictive_Modelling_and_Analytics_for_Diabetes_using_a_Machine_Learning_Approach )

Round 2

Reviewer 1 Report

All the necessary changed have been made. I am recommending to accept the manuscript in present form.